# The role of dispersal and school attendance on reproductive dynamics in small, dispersed populations: *Choyeros* of Baja California Sur, Mexico

**Shane J. Macfarlan** [1,2,3]*, **Ryan Schacht** [4], **Eric Schniter**[5,6], **Juan José Garcia**[7], **Diego Guevara Beltran**[8], **Jory Lerback**[9]

**1** Department of Anthropology, University of Utah, Salt Lake City, UT, United States of America, **2** Center for Latin American Studies, University of Utah, Salt Lake City, UT, United States of America, **3** Global Change and Sustainability Center, University of Utah, Salt Lake City, UT, United States of America, **4** Department of Anthropology, East Carolina University, Greenville, NC, United States of America, **5** Economic Science Institute, Chapman University, Orange, CA, United States of America, **6** School of Humanities & Social Science, Salt Lake Community College, Salt Lake City, UT, United States of America, **7** Department of Anthropology, California State University Fullerton, Fullerton, CA, United States of America, **8** Department of Psychology, Arizona State University, Tempe, AZ, United States of America, **9** Department of Geology & Geophysics, University of Utah, Salt Lake City, UT, United States of America

* shane.macfarlan@anthro.utah.edu

**Data Availability Statement:** All relevant data are within the paper and its Supporting Information files.

## Abstract

Individuals from small populations face challenges to initiating reproduction because stochastic demographic processes create local mate scarcity. In response, flexible dispersal patterns that facilitate the movement of individuals across groups have been argued to reduce mate search costs and inbreeding depression. Furthermore, factors that aggregate dispersed peoples, such as rural schools, could lower mate search costs through expansion of mating markets. However, research suggests that dispersal and school attendance are costly to fertility, causing individuals to delay marriage and reproduction. Here, we investigate the role of dispersal and school attendance on marriage and reproductive outcomes using a sample of 54 married couples from four small, dispersed ranching communities in Baja California Sur, Mexico. Our analyses yield three sets of results that challenge conventional expectations. First, we find no evidence that dispersal is associated with later age at marriage or first reproduction for women. For men, dispersal is associated with younger ages of marriage than those who stay in their natal area. Second, in contrast to research suggesting that dispersal decreases inbreeding, we find that female dispersal is associated with an increase in genetic relatedness among marriage partners. This finding suggests that human dispersal promotes female social support from genetic kin in novel locales for raising offspring. Third, counter to typical results on the role of education on reproductive timing, school attendance is associated with younger age at marriage for men and younger age at first birth for women. While we temper causal interpretations and claims of generalizability beyond our study site given our small sample sizes (a feature of small populations), we nonetheless argue that factors like dispersal and school attendance, which are typically

**Funding:** SJM and ES (HJ-099R-17) National Geographic Society Research and Exploration Grant. SJM (subcontractor 1743019) National Science Foundation IBSS-L (PI Koster) https://www.nsf.gov/awardsearch/showAward?AWD_ID=1743019&HistoricalAwards=false. The funders had no role in study design, data collection and analysis, decision to publish, or preparation of the manuscript. In addition, this research was supported through the following sources: Funding Incentive Seed Grant (University of Utah), the Center for Latin American Studies (University of Utah), Society, Water, & Climate Seed Grant (University of Utah), Nexus Pilot Grant (University of Utah), Economic Science Institute (Chapman University), and the Division of Anthropology (California State University Fullerton).

**Competing interests:** The authors have declared that no competing interests exist.

associated with delayed reproduction in large population, may actually lower mate search costs in small, dispersed populations with minimal access to labor markets.

## Introduction

Small populations are prone to stochastic demographic processes that can lead to a host of challenges for initiating reproduction [1–4]. For example, chance sex-biases in interannual birth or death rates can lead to highly variable adult sex ratios across time and place [5–8]. The consequences of partner unavailability are further intensified in small communities located in areas of low population densities (e.g., rural communities). Consequently, finding a mate who is sexually available and of low genetic relatedness can be difficult, if not impossible, within small groups. Revealing how individuals in small communities overcome the challenges of mate search and acquisition is important for understanding individual, family, and population health because partner availability has been shown to impact patterns of sexual risk-taking [9,10], pair-bond stability [11,12], parental investment [13], and violence [14–16]. Moreover, given that small communities (i.e., groups between 10–150 individuals; [17–19]) were typical throughout most of human history [20–22], understanding how individuals and groups cope with mate search will aid in reconstructing evolutionary patterns of human sociality [e.g. 8,23].

Local mate scarcity and other consequences of small populations are not unique to humans [4]. One solution across the animal kingdom to avoid inbreeding depression is dispersal [24–28]. The typical pattern across most sexually reproducing species is sex-specific dispersal [29–31]. However, while dispersal can assist in expanding an individual's mating market, it can also be costly. This has been well-documented across a wide range of animal species and includes, for example, loss of body mass, elevated stress levels, compromised immune functioning, and delayed reproduction [32–34]. Dispersal costs are multivariate, but for group-living species they are typically tied to reliance on social relationships for reproductive opportunities, protection, and resource provisioning [34]. Because dispersal requires individuals be removed from existing social support networks (either voluntarily or involuntarily) and travel to novel locales where they lack kin or alliance partners, individuals who disperse are expected to incur costs.

However, humans are distinctive from other group-living species in a number of ways. One hallmark characteristic is the dynamic nature of our social organization within and between groups [35–37]. Dispersal from the natal area is commonly observed cross-culturally, yet which sex leaves is highly variable [21,38]. While this flexibility has been argued by some to minimize the costs to dispersal (e.g., who disperses is responsive to local mate scarcity [8]), researchers commonly report costs sustained by the dispersing sex. For example, using historical datasets, delayed age at marriage and first reproduction among those who disperse are often found [39–42]. These costs, though, are typically reported for larger populations where partners are often available in the source population and so those who disperse do so under less than ideal conditions (e.g., local resource scarcity or economic hardship [43,44]. However, in small communities, few marriage options exist in the natal area and so dispersal may serve to hasten, rather than delay, the initiation of reproduction [*sensu* 45,46].

A second characteristic of the human niche is our ability to form long-term cooperative relationships to achieve individual and group-level goals [21,23]. While these high levels of cooperation are promoted by a number of mechanisms [47], genetic kinship appears to be foundational to human social organization [23,35,48–50] and especially relevant in modern

small-scale societies [36,37,51]. For example, as humans are a cooperatively breeding species [52–56], women are able to maintain multiple dependent offspring as well as a rapid reproductive pace compared to other apes because they receive allomaternal support, often from maternal genetic kin [48,56–58]. Furthermore, in many contexts, such as labor or social support, male-male coalitions appear to be predicated on genetic kinship [36,37,49,51,59]. However, this raises concerns for the costs of dispersal, because it risks removing individuals from vital support networks composed of genetic kin. Humans appear to have worked around this by way of a cross-culturally common pattern whereby social institutions facilitate the movement of genetic kin between communities through marriage to cousins or to more distantly related kin [60,61]. In this way, individuals who disperse, enter into households and communities where they are already embedded into local social support networks comprised of genetic kin [62]. Accordingly (and, to some extent, counterintuitively), human dispersal may occur in such a way not to dampen inbreeding depression per se but, rather, to *promote inbreeding*. We suggest that this pattern arises because social support networks composed of genetic kin are such an important determinant for achieving improved economic and/or reproductive outcomes in humans.

As outlined above, the role of dispersal on reproductive outcomes remains an open question. We target this lack of consensus in the literature by way of reproductive outcomes across several small communities where few to no partners are available locally because of the demographic realities of small groups. Accordingly, we expect that dispersal serves to lower the costs to initiating reproduction compared to those who stay in their natal area where partners are rare. Moreover, we expect that men and women who disperse will be more likely to marry kin than those who stay in their natal area.

Important to consider are additional drivers of human fertility that differ from other organisms, as well as our past, due to contemporary socioecological environments that are embedded within a larger regional or national economy [63,64]. State-sponsored structures are typically in place to support individual contribution to and competitiveness in economic markets [65]. While variable across place, a nearly ubiquitous feature of contemporary socioecological environments is access to formal education through compulsory school attendance [66]. A typical, yet seemingly unintended, consequence is that ages of marriage increase and fertility rates decrease with attending school [67,68]. One central explanatory framework for this relationship is Embodied Capital Theory [69]. For example, all individuals face reproductive tradeoffs and, with respect to education, individuals curtail investment in reproduction to increase their or their children's success in future reproduction by way of achieving competency within the economic environment. However, within small rural populations, formal education may be less useful for access to economic markets [70] and instead may be used to expand mating markets outside of one's natal area and gain access to state-sponsored resources (e.g. food, medical services). Thus, in addition to exploring the consequences and patterning of dispersal, we also seek to assess the role of school attendance on reproductive outcomes in small communities.

Below we examine the following questions about reproductive dynamics across four small communities from Baja California Sur, Mexico: 1) Do dispersal and school attendance decrease the age at marriage for men and women? 2) Does dispersal increase marriage partner genetic relatedness? 3) Do dispersal and education decrease the age at first reproduction for women? In sum, while dispersal and school attendance have typically been portrayed as costly to humans by way of fertility, this is likely relevant to large populations and not those typical of many contemporary small-scale groups, as well as for much of human history. Moreover, while dispersal is typically argued as a strategy for minimizing inbreeding depression,

marrying kin outside of the local group may instead serve to minimize reproductive costs by leveraging the support of pre-established social networks composed of genetic kin.

## Materials and methods

### Study site

The *Sierra de La Giganta* (hereafter "*Giganta*") is Baja California Sur, Mexico's largest mountain range, spanning ~150 km along a NW-SE axis with a total surface area of ~7,400 square kilometers [71]. Biogeographically, the *Giganta* is characterized as Sonoran Desert [72] with scrubland vegetation dominated by woody legumes [e.g., Palmer Mesquite (*Prosopis palmeri*)], columnar cacti [e.g., Organ Pipe Cactus (*Stenocereus thurberi*)], and palm-lined oases [e.g., Mexican Fan Palm (*Washingtonia robusta*)] [72]. It has a hot, arid climate (Köppen-Geiger BWh) with most of its precipitation (~200 mm annually) occurring during the mid-summer to early fall (July-September) as monsoonal rains [72]. Although the range is situated more closely to the Gulf of California along its eastern escarpment, it slopes towards the west, producing a number of intermittent-stream drainages that terminate at the Pacific Ocean [73]. It is along these drainages that perennial wetlands are present, typically as springs [74,75], which represent the only permanent source of fresh water in this desert environment and make sedentary human life possible.

Although humans have occupied the *Giganta* for at least the last 4000 years [76,77] people of Euro-American descent began permanently occupying the region in 1697 AD following the establishment of the Jesuit mission of Loreto [78,79]. In order to more successfully colonize the peninsula, the Jesuits brought with them individuals and families to act as soldiers, metal smiths, leather workers, cattle herders, farmers, and teachers [80]. These early settlers, along with three additional waves of colonists who entered the peninsula following the Jesuit expulsion (1768 AD), Mexican Independence (1821 AD), and the *Porfiriato* Period (1875–1910 AD), form the genealogic roots of many modern Baja California peoples, including the *Choyero* ranching communities of the *Giganta* [42,81,82]. Historical demographic research suggests that male-biased dispersal was typical during the 19th century [42].

Currently, approximately 4000 people reside across the *Giganta* [83] resulting in a population density of about one person per two square kilometers. Households are predominantly located within valleys on flat-lands above dry riverbeds near springs. Four communities located in the southern *Giganta* are the focus of this study. While not a closed population, they were chosen because they represent a large segment of the mating pool for most residents in the area. Three communities are located within the most upstream sections of the Santa Rita watershed (Santa Maria de Toris, San Pedro de La Presa, and La Higuera), while the fourth is located in the most upstream section of the Las Pocitas-San Hilario watershed (La Soledad; Fig 1). These communities lack infrastructure development such as piped water, sanitation, electricity, and paved roads [83–85].

Their primary form of subsistence relies on animal husbandry, with an emphasis on meat and cheese production for household consumption as well as for sale at local and regional markets. Households tend to specialize in either goat or cattle production, but also maintain other livestock for domestic consumption (e.g., sheep and chickens) or transportation (e.g., horses, donkeys, and mules). Some men and women additionally contribute to household income by way of artisanal crafts they make and sell at local and regional markets. More generally, however, households supplement their diet through a government-sponsored food program, as well as through purchased food from urban markets (e.g., Ciudad Constitución, La Paz, Loreto). A number of ranches maintain *huertas* and/or *jardins*—men's and women's gardens, respectively, located near homes. While *huertas* provide comestible resources for the

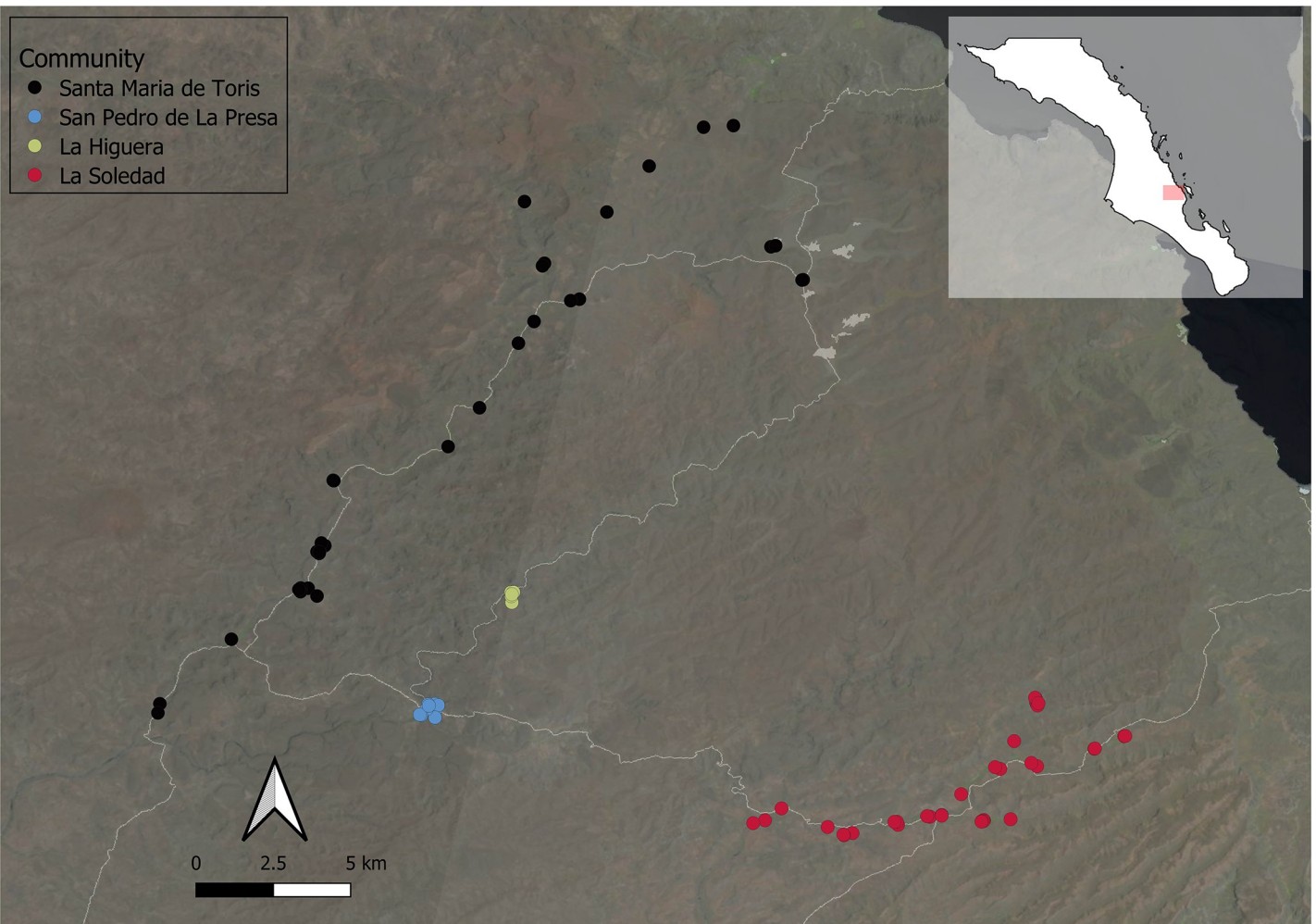

**Fig 1. Location of the four *Choyero* ranching communities.** Each dot represents a single household. White lines represent dirt roads.

household and feed for domestic livestock (e.g., sorghum), they also serve as important repositories for Jesuit mission era crops such as figs, mangoes, limes, and oranges [86]. *Jardins*, on the other hand, serve important household functions such as shade and medicine. Land tenure is mixed, with some households located on private property, others on common-pool land units (*ejidos*), and still others which lack clear land title and therefore exist on contested lands. The predominant religion is Catholicism.

Based on the 2015 Mexican intercensal, Baja California Sur (BCS) has the second smallest population (745,601 people) out of Mexico's 32 states but represents the seventh largest state by surficial land area (73,909 square kilometers) [87]. As such, it has the lowest population density and represents one of the most rural states in Mexico [87]. In an effort to improve the social wellbeing of its rural populace, the BCS state government has promoted education through the *Coordinación Estatal de Albergues Escolares*. This education program, which is unique to BCS, provides rural communities access to primary and/or secondary education through the placement of schools, cafeterias, dormitories, teachers, and social welfare officers in rural locations. Currently, 31 *albergues escolares* exist in BCS [88]. Rural children are brought to the school for five days a week and then return home every weekend. Although the

*albergues escolares* program has existed in BCS for over sixty years, within the study site the program is comparatively new, with two *albergues* established in 1969 and 1980 in La Soledad and Santa Maria de Toris, respectively–two of the four communities. Additionally, a learning center was initiated in 2003 in the community of La Higuera (a third of the four communities) that is composed of a single, one-room structure for one teacher to provide primary education exclusively to the children of this community. This educational facility is not associated with the *albergues escolares* program. Although education has been on the rise among rural BCS families, there is variability in attendance, attrition rates, and educational outcomes [83]. Because these rural communities lack public transportation to schools, children who reside distantly from schools less regularly attend than those who live immediately adjacent to them. Socioeconomic factors too play a role in attendance and attrition. The families of children who must be driven to school may lack the resources necessary to pay for gasoline and vehicle maintenance, while others may lack the funds to pay for educational fees. Additionally, children are often engaged in domestic labor, causing some to either fail to complete their education, or to need to forgo it altogether.

## Data

**Ethics.** Permission to conduct this research was obtained through the University of Utah Institutional Review Board (IRB # 00083096), as well through signed written agreements with official representatives ("*subdelegados*") from the four communities. In accordance with each oversight body, consent was obtained from all head of households to conduct research, which was recorded by the lead investigator (SJM) at the time of the interview. Because not all participants could read or write, consent to participate was established verbally following a description of the project.

**Community census.** Community size, household composition, and demographic structure were obtained via a series of interviews with heads of households from 2015 to 2018. Across the four communities there were 90 households and 295 individuals in total (Fig 1; Table 1). Community size has a bimodal distribution (range: 28–123) (Fig 2) and all

**Table 1. Descriptive statistics associated with census and marriage data.**

| | Yes | No | n | Mean (SD) | Median | Min/Max |
|---|---|---|---|---|---|---|
| | | | Census Data | | | |
| **Household Size: Santa Maria de Toris** | - | - | 37 | 3.3 (1.3) | 3 | 1/7 |
| **Household Size: San Pedro de La Presa** | - | - | 10 | 2.8 (1.1) | 3 | 1/5 |
| **Household Size: La Higuera** | - | - | 7 | 4.6 (0.8) | 5 | 3/5 |
| **Household Size: La Soledad** | - | - | 36 | 3.1 (1.7) | 3 | 1/6 |
| | | | Marriage Data | | | |
| **Couple Genetic Relatedness** | - | - | 52 | 0.017 (0.04) | 0 | 0/.156 |
| **Year of Marriage** | - | - | 53 | 1990 (15) | 1989 | 1961/2016 |
| **Groom Age at 1st Marriage** | - | - | 50 | 25.9 (6) | 25 | 15/46 |
| **Bride Age at 1st Marriage** | - | - | 53 | 21.6 (6) | 20 | 14/40 |
| **Spousal Age Difference (Groom Age–Bride Age)** | - | - | 50 | 5 (6.6) | 5 | -9/21 |
| **Bride Age at 1st Birth** | - | - | 50 | 22.7 (5) | 22 | 16/35 |
| **Years of Education** | - | - | 101 | 4.5 (3) | 5 | 0/12 |
| **Groom Ever Attended School** | 38 | 13 | 51 | - | - | - |
| **Bride Ever Attended School** | 42 | 11 | 53 | - | - | - |
| **Groom Dispersed** | 18 | 35 | 53 | - | - | - |
| **Bride Dispersed** | 31 | 22 | 53 | - | - | - |

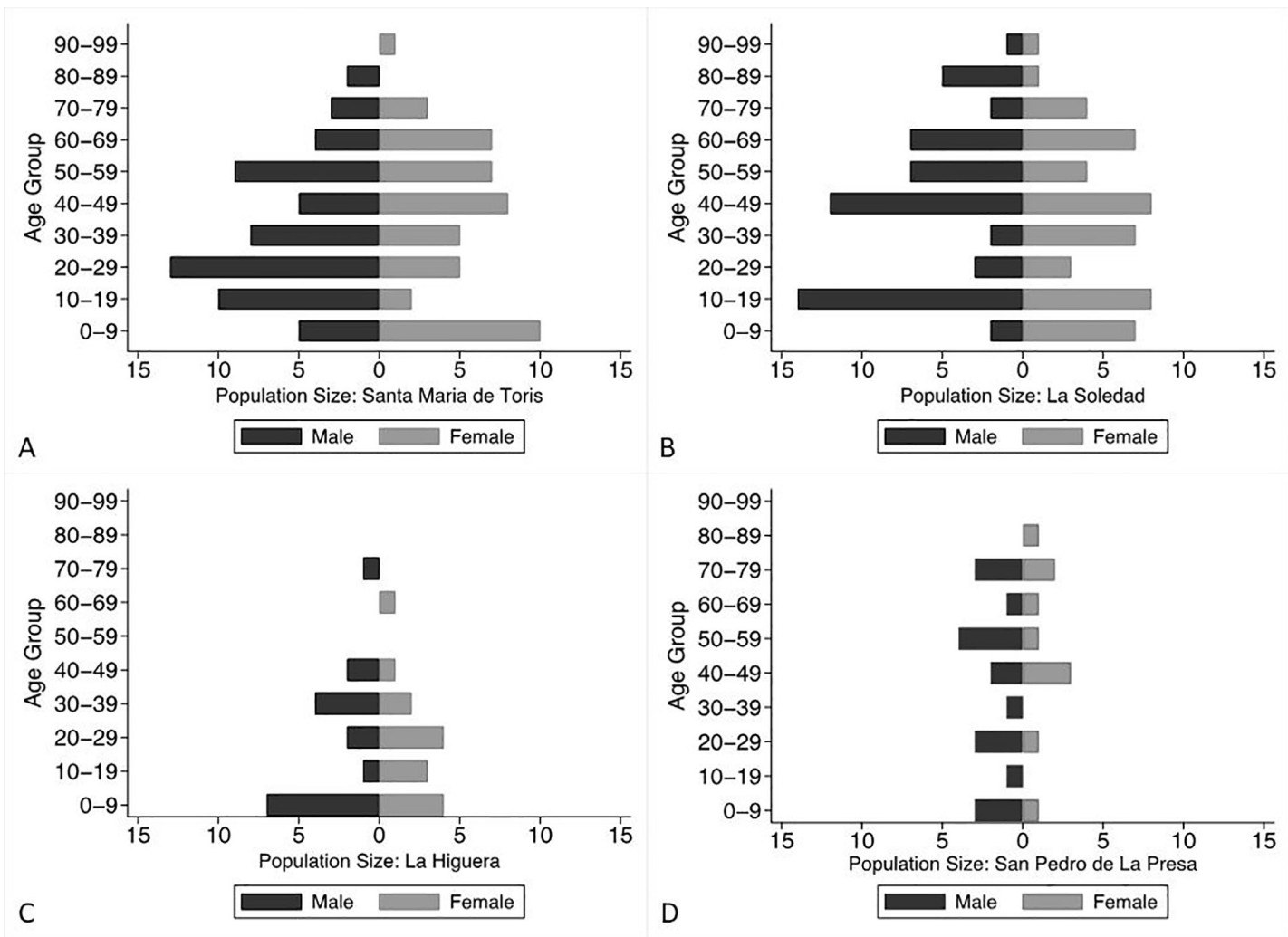

**Fig 2. Age and sex distributions for the four _Choyero_ ranching communities.**

communities demonstrate male-biased sex ratios at the time of data collection (mean = 1.2; range: 1.07 to 1.8). Ethnographic interviews suggest this bias is largely driven by female dispersal to both other ranching communities as well as to urban environments.

**Dispersal, marriage, and reproductive data.** Detailed dispersal, marriage, and reproductive data were collected from the heads of 54 households (60% of all households across the four communities) and is available as supporting information (S1 Data). Interviews were conducted in Spanish and included questions about the timing of marriage (age and year of marriage) and first reproduction, as well as the social context in which individuals met their mates. Dispersal was operationalized by determining whether an individual currently resided in the community in which they were born and raised Table 1. Both men and women disperse; however, a Chi square test shows that women were statistically more likely to disperse from their natal community ($X^2$ = 6.4; n = 106; p = .01) as reflected in the male biased sex-ratio. Because of their reliance on livestock production, ranches require male head of households and their sons to care for livestock. Furthermore, habitat saturation has limited men's ability to establish independent households. Accordingly, after marriage, sons typically reside in dwellings

immediately adjacent to their parents' home to allow for mutual aid due to subsistence practices and resource limitation.

Of the 54 households, 52 provided information on the social context in which they had met their partners. Seventeen couples met at a rural school, fifteen met at religious or civic festivals, twelve met while visiting other communities (often because they were looking for missing livestock), and eight met through mutual relatives.

Of the 108 married individuals interviewed, 104 had information on educational attainment. Average education attainment was four years (min/max = 0/12 years) and no difference existed between men and women in the number of school years attended (t = 0.6; d.f. = 102; p = 0.3). A relationship existed between education and dispersal, such that individuals who had attended school were less likely to disperse relative to those with no education ($X^2$ = 4.8; p = .03; n = 104; Natal-Education n = 47; Dispersed-Education n = 33; Natal-No Education n = 8; Dispersed-No Education n = 16). Furthermore, OLS regression, examining the relationship between year of birth and years of education, shows that education achievement has increased over time in this sample (B = 0.13; p < .001; n = 101).

**Kinship.**   As is customary in anthropological kinship studies [e.g., 89], genetic kinship data were obtained via a series of genealogical interviews with all heads of households across the four communities between 2015 and 2018. The database contains information on 1032 individuals born between the late 1700's and 2018. Genetic relatedness was calculated using the software *Descent* [90]. Average genealogic depth is three-and-a-half generations (range: 2–6 generations) for the 54 interviewed couples. Thirteen couples (25 percent) were genetically related with a maximum genetic relatedness of 0.156 (i.e., first-cousins).

**Analysis.**   Our analytical models are performed using STATA/IC 15.0 [91]. We apply two classes of models depending on the outcome variable. For the outcomes "groom age at first marriage", "bride age at first marriage", and "bride age at first birth", we employ Generalized Estimating Equations (GEE) using the xtgee command. This approach allows us to estimate regression coefficients for fixed effects, while simultaneously accounting for the nested structure of our data. In each analysis, we nest the data at the level of the community using an independent correlation structure and employ Robust Standard Errors (RSE). Furthermore, because these outcome variables are continuous, we employ a Gaussian distribution and identity link function. For the outcome, "marriage couple genetic relatedness", we employ Fractional Regression, using the fracreg command. This approach allows us to estimate regression coefficients for an outcome variable that ranges between zero and one. To account for the clustered nature of the data around communities, we use Clustered Robust Standard Errors and employ a Logit distribution. The following variables are included in our analyses: **Outcomes** 1) age at marriage (a continuous variable), 2) marriage partner genetic relatedness (a fractional variable), 3) age at first birth (a continuous variable); **Predictors** 4) dispersal (a binary variable: 1 = dispersed; 0 = did not disperse), 5) education (a binary variable: 1 = attended rural school; 0 = did not attend rural school); **Control** 6) year of marriage (a continuous variable).

Before moving to our analysis, we would like to make clear some limitations to our study design. First, our analytic models do not allow us to isolate causality. For example, for each individual, we do not know the entirety of the pool of potential mates available to them at the time that they were married. As a result, we are careful to interpret our findings as associations between variables. Second, our dataset does not include information on *why* some individuals obtained a particular level of education or *why* they did or did not dispersed. Thus, factors that promote some to go to school or disperse, such as wealth or parental education, are unable to be accounted for in our statistical models. Lastly, while a feature of small communities, because our sample sizes are small, we are careful when interpreting our findings relative to other populations.

**Table 2. GEE Gaussian regression models explaining age at first marriage.**

| | b (RSE) | Z | p |
|---|---|---|---|
| Male Model[1] | | | |
| Dispersed (1 = Yes; 0 = No) | -2.4 (0.09) | -27.2 | < .001 |
| Ever Attended School (1 = Yes; 0 = No) | -5.02 (1.2) | -6.2 | < .001 |
| Year of Marriage | 0.06 (0.07) | 1.0 | .33 |
| Constant | -97 (130) | -0.7 | .46 |
| Female Model[2] | | | |
| Dispersed (1 = Yes; 0 = No) | -1.5 (1.5) | -0.9 | .34 |
| Ever Attended School (1 = Yes; 0 = No) | -4.7 (2.2) | -2.1 | .04 |
| Year of Marriage | -0.01 (0.03) | -0.2 | .84 |
| Constant | 37 (52) | 0.7 | .48 |

[1] Wald $X^2$ = 1024.8; n-groups = 4; n-observations = 48; p < .001.

[2] Wald $X^2$ 4.7; n-groups = 4; n-observations = 52; p = .2.

## Results

### What are the effects of dispersal and education on age at first marriage?

To determine whether dispersal and education impact marriage outcomes, we employ sex-specific analyses. Furthermore, we include year of marriage to control for secular trends in both models. For men, we find that both dispersal (b = -2.4, p < .001) and having ever attended school (b = -5.0, p< .001) is significantly associated with a decrease in the age at marriage (Table 2). For women, we find that dispersal (b = -1.5, p = .3) has no association with age at marriage, while attending school (b = -4.7, p = .04) is associated with younger marriage ages (Table 2).

### What is the effect of dispersal on marriage partner genetic relatedness?

To test how dispersal affects marriage partner genetic relatedness, we perform an analysis that examines both male and female dispersal simultaneously and include year of marriage to control for secular trends. We find that male dispersal has no significant relationship with marriage partner genetic relatedness (Odds Ratio = 1.4; p = .72), while for women, dispersal is significantly associated with an increase in the odds of marrying a genetic relative (Odds Ratio = 15.7; p = .01) Table 3 (Fig 3).

### What are the effects of dispersal and school attendance on age at first birth?

Research typically indicates that dispersal and school attendance can cause individuals to delay reproduction. However, our analyses above suggest that neither were associated with an

**Table 3. Fractional regression model1 explaining marriage couple genetic relatedness.**

| | OR (CRSE)[2] | z | p |
|---|---|---|---|
| Female Dispersed | 15.7 (16.9) | 2.6 | .01 |
| Male Dispersed | 1.4 (1.3) | 0.4 | .72 |
| Year of Marriage | 0.9 (0.02) | -0.3 | .80 |
| Constant | 15 (565) | 0.1 | .94 |

[1] Wald $X^2$ = 23.9; n-groups = 4; n-observations = 51; p < .001.

[2] Clustered Robust Standard Errors.

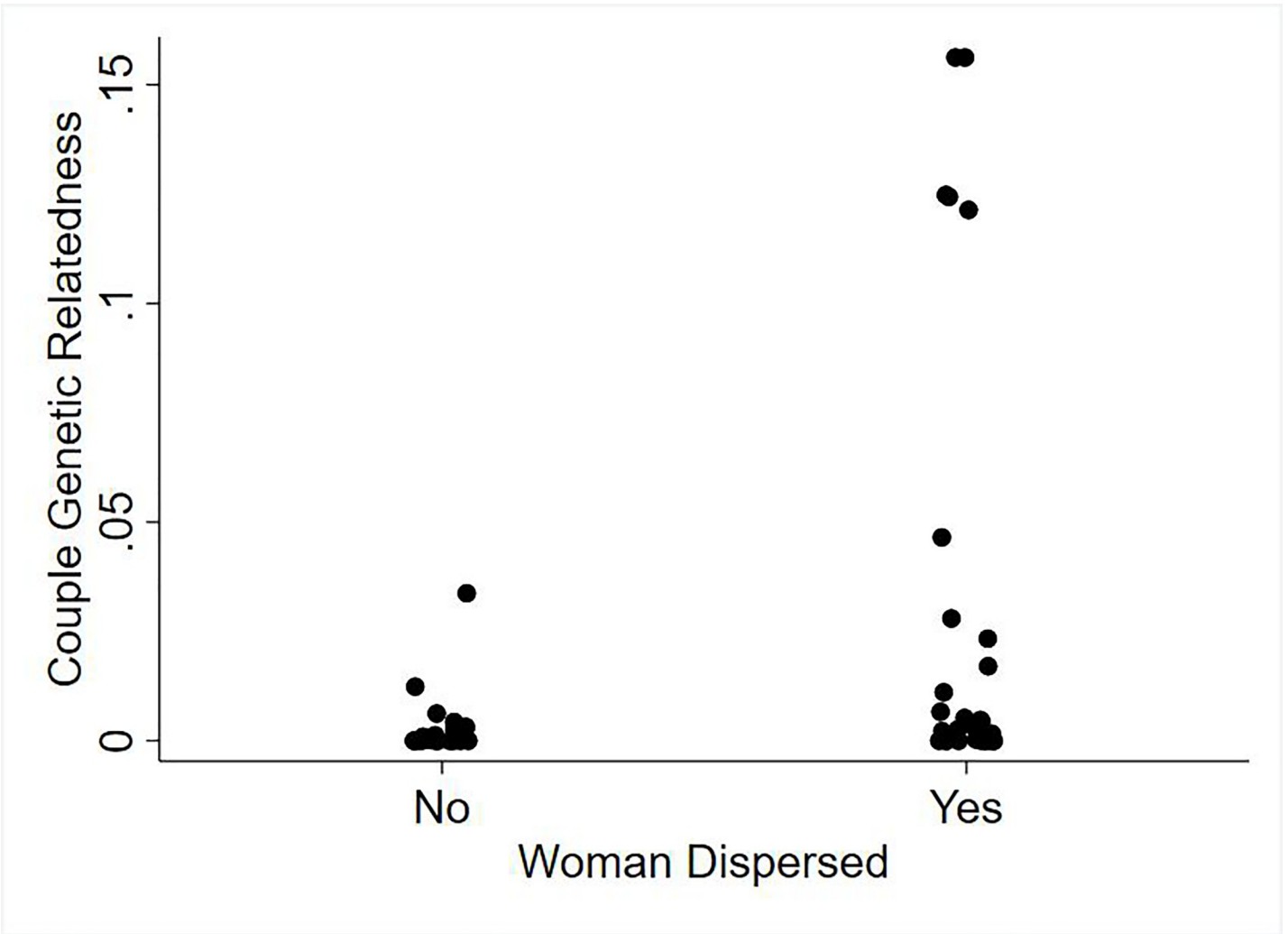

**Fig 3. The relationship between female dispersal and marriage couple genetic relatedness.**

increase in female age at marriage, suggesting that these factors have dissimilar impacts in small, dispersed communities. As such, they too could have contrasting effects on female age at first reproduction. Because of the known statistical correlations between age at marriage and first birth [92], we control for age at marriage. We find that education (b = -1.9; p < .001), but not dispersal (b = -0.07; p = .9), is significantly associated with a decrease in the age at first birth (Table 4).

**Table 4. GEE Gaussian regression model[1] explaining female age at first birth.**

|  | b (RSE) | Z | P |
|---|---|---|---|
| Female Dispersed (1 = Yes; 0 = No) | -0.07 (0.8) | -0.9 | .93 |
| Ever Attended School (1 = Yes; 0 = No) | -1.9 (0.3) | -5.8 | < .001 |
| Female Age at Marriage | 0.8 (0.02) | 36.7 | < .001 |
| Constant | 7.8 (0.8) | 9.6 | < .001 |

[1]Wald $X^2$ = 1875.8; n-groups = 4; n-observations = 49; p < .001.

## Discussion

Our analyses among men and women living across four small, rural communities yield three sets of results that challenge conventional expectations regarding consequences to dispersal and education. First, dispersal may play an important role in *minimizing* reproductive costs. Contrary to previous research on the topic [39–41], we find no evidence that dispersal is associated with an increase in the age at marriage or first reproduction in women. And among men, dispersal is associated with younger ages of marriage than those who stay in their natal area. Second, while dispersal has been presented as a way to manage inbreeding depression in one's natal group [93], we find that female dispersal is statistically associated with an increased likelihood of marrying genetic kin. That is, female dispersal may increase rather than decrease genetic relatedness among marriage partners. Third, counter to typical results for the role of education on reproductive timing [63–65], attending school is associated with a lower age at marriage for men and lower age at first birth for women. Below we offer interpretations of our findings and their possible applications to the literature on small populations across the social and biological sciences.

Taken together, our analyses demonstrate that factors typically associated with individuals delaying reproduction in large populations may actually accelerate marriage and reproduction in small, dispersed communities. Given that small communities are prone to demographic processes that can lead to local mate unavailability [1–3], dispersal and school attendance can serve to expand mating markets and lower partner search costs. As such, reproductive dynamics should be expected to vary by community size (as is commonly reported in the nonhuman animal literature). For example, those who disperse in large populations typically do so not because of a lack of local mates, but often due to local resource scarcity and/or economic hardship [41,45]. However, within the small communities presented here, dispersal allows men to secure a mate more quickly than those who stay home and face local mate unavailability. While dispersal is often presented as being either uniformly negative or positive on individual outcomes, a more nuanced approach is likely appropriate given that decisions to migrate are conditioned by environmental and/or individual-level variability, such as local mate scarcity, habitat suitability and saturation, and kinship institutions. We plan to target future research on the multiple motivations for and consequences of dispersal on individual outcomes.

We also find that dispersal has a sex-specific association with marriage partner genetic relatedness. This implies sex differences in the structure and function of social support networks, as well as the potential for parent-offspring conflicts in mating decisions [*sensu* 62]. For example, men are less likely to disperse than women. Why? Across these ranching communities, males spend substantial parts of their day covering large areas within the desert-mountains herding and caring for livestock. Given that this work is done by fathers and their sons, and is necessary for household functioning, parents desire sons to stay at home. Women, on the other hand, are more likely to disperse than males and, when they disperse, are more likely to marry males who are genetic kin. Why? Child health and wellbeing is clearly tied to genetic kin support across societies. It is well-documented that offspring outcomes are improved when mothers have support from family members, especially maternal kin [reviewed in 57]. Thus, to dampen the costs of dispersal where males are the philopatric sex, women may target marrying genetic kin from distant communities as a means to re-establish social support outside of their natal area. This may provide young mothers with the alloparental support necessary for rearing children in novel social environments. Thus, we argue that the linkage between dispersal and marriage to genetic kin represents a possible mechanism to deal with both local mate scarcity and the needs of a hyper-cooperative species that relies on genetic kinship for rearing offspring [48,50,52–54,62].

In contrast to much research on the effect of education on the initiation of reproduction [65–69], we find that school attendance is associated with an acceleration of the onset of marriage and reproduction in these rural ranching communities. Rather than interpreting this finding as contradictory to previous research, we instead highlight the nature of education in this rural context. Ethnographic interviews with ranchers suggest that although education improves academic skills acquisition, school attendance largely centers on improving opportunities for socialization as well as gaining access to state sponsored resources, such as meal provisioning. Due to both the limited quality of rural education programs, as well as the lack of easy access to well-developed labor markets in cities, rural ranchers are at a competitive disadvantage for acquiring jobs that demand skills gained through compulsory education. Instead, school attendance appears to assist people with locating social partners by aggregating children across large geographic distances that are often difficult to traverse. This phenomenon is likely typical of other rural, dispersed, economically transitioning populations where state sponsored institutions allow individuals to aggregate for extended periods of time, but labor markets are underdeveloped or distant [94].

Before concluding we would like to highlight some limitations of our work. First, we are unable to determine causality due to our methodological approach. While the data collection protocol allows us to understand current community structure and mate availability, it does not allow us to reconstruct these items in the past prior to marriage. Second, with our data, we cannot determine why some individuals dispersed or went to school while others did not. The factors that influence who disperses and attends school (e.g. household wealth, parental education, and distance to schools), are potential confounds that may impact marriage and reproductive outcomes. While variability in these factors plays an important role in education attainment and dispersal in many groups (e.g. rural-to-urban migration or in international settings) [39,45], virtually all of the dispersal dynamics reported here occur within the ranching communities of the *Giganta*, where individuals are moving from one rural ranching community to another. As such, it remains open as to how factors motivating dispersal operate in small, rural, dispersed communities. Lastly, our sample sizes are small given the realities of working with small populations and this impacts our ability to make strong claims of generalizability across place. However, while our sample size is small, our ethnographic description and detailed information for each individual provides a rich tapestry from which to understand and interpret results. Future work could move the literature forward by both addressing these limitations and applying the insights detailed here to other locales to better understand reproductive dynamics across small populations.

In conclusion, small communities located within meta-populations of low population density present considerable challenges for initiating reproduction. While dispersal and school attendance have typically been portrayed as costly to humans by way of fertility, this expectation is likely relevant to large and not small populations like those presented here. We instead find that dispersal and education both lower reproductive costs and allow people to initiate reproduction at earlier ages. Moreover, while dispersal is typically argued as a strategy for minimizing inbreeding depression, we find that women are more likely to marry kin if they disperse. Marrying kin when dispersing from the natal area may serve to minimize reproductive costs by way of integrating women into pre-established social networks necessary to aid in raising altricial young. In sum, small communities characterize the social structure for much of human evolution as well as for the world's rural people today. Accordingly, predictions and findings from contemporary groups with large populations regarding fertility may not be scalable to small-scale communities and thus require population size to be considered as an important component to the selective area.

## Supporting information

**S1 Data.**
(XLSX)

## Acknowledgments

The authors wish to thank the gracious support and hospitality of the people of Santa Maria de Toris, La Higuera, San Pedro de La Presa, and La Soledad, with a special thanks to the Amador and Bibo families. We also thank the editor, David Lawson, as well as Thomas Kraft and one anonymous reviewer for their insightful reviews and commentary that have improved this manuscript.

## Author Contributions

**Conceptualization:** Shane J. Macfarlan, Ryan Schacht.

**Data curation:** Shane J. Macfarlan.

**Formal analysis:** Shane J. Macfarlan.

**Funding acquisition:** Shane J. Macfarlan, Eric Schniter.

**Investigation:** Shane J. Macfarlan, Eric Schniter, Juan José Garcia, Diego Guevara Beltran.

**Methodology:** Shane J. Macfarlan.

**Project administration:** Shane J. Macfarlan.

**Resources:** Shane J. Macfarlan.

**Software:** Shane J. Macfarlan.

**Supervision:** Shane J. Macfarlan.

**Visualization:** Shane J. Macfarlan, Jory Lerback.

**Writing – original draft:** Shane J. Macfarlan, Ryan Schacht, Eric Schniter.

**Writing – review & editing:** Shane J. Macfarlan, Ryan Schacht, Eric Schniter.

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
