## [Decision Letter · Decision Letter 0]

3 Jul 2020

PONE-D-20-06987

The role of dispersal and school attendance on reproductive dynamics in small, dispersed populations: Choyeros of Baja California Sur, Mexico

PLOS ONE

Dear Dr. Macfarlan,

Thank you for submitting your manuscript to PLOS ONE. After careful consideration, we feel that it has merit but does not fully meet PLOS ONE’s publication criteria as it currently stands. Therefore, we invite you to submit a revised version of the manuscript that addresses the points raised during the review process.

Editor comments:

There is a lot to like about this manuscript! Thank you for opportunity to consider your work. Two reviewers have provided supportive criticism, and agree this is well-written engaging scholarship. However, some concerns remain. While mostly minor concerns, I would like you to respond carefully to each suggestion point by point in your reply. Please note that Reviewer 1 also made some comments on an attached PDF for you to consider. In addition, I request that you be especially attentive to the following concerns. All of which I consider requirements before the manuscript would be acceptable for publication:

1. Eliminate causal language when your results only imply statistical relationships. This applies throughout the manuscript. For example, in the abstract: “For men, dispersal results in younger ages of marriage” – should become “For men, dispersal is associated with younger ages of marriage”.  

2. While you do discuss limitations at the end of the manuscript. I would like to see a more dedicated, upfront acknowledgement and discussion of the limitations of your methodology specifically with respect to isolating causality. My concerns here are not so much that causality might be reversed, but rather that the associations you report are very likely also associated with confounding 3rd variables. There is a general sense in which you treat going to school and dispersal as almost experimental factors, but a large body of demographic research tells us that migrants are rarely comparable to those who do not migrate. Similarly, those that go to school likely come from families of distinct socioeconomic backgrounds that those who do not. Therefore your manuscript must more clearly address which individuals are most likely to disperse and go to school and discuss the role of potential confounds. The reality is that while you have a lot of ethnographic knowledge to draw on, and a very sophisticated grasp of the theory and past literature- the statistical models are very simplistic and the analysis dangerously underpowered - especially by the standards of demography. I realize these limitations are balanced by the novelty of the data and fieldwork commitments, but they need to be more clearly acknowledged. 

3. A concise mention of these limitations should be made in the abstract itself. The abstract should also mention the sample size.

4. Linking to the above points, reading the manuscript I was not entirely clear of the extent to which dispersal and education are overlapping phenomena for men and women. How many folks go to school but don't disperse and vice-versa. This needs to be made more clear. 

We look forward to receiving your revised manuscript.

Kind regards,

David W Lawson

Academic Editor

PLOS ONE

Journal Requirements:

2. During our internal checks, the in-house editorial staff noted that you conducted research or obtained samples in another country.

Please check the relevant national regulations and laws applying to foreign researchers and state whether you obtained the required permits and approvals. Please address this in your ethics statement in both the manuscript and submission information.

3. Please change "female” or "male" to "woman” or "man" as appropriate, when used as a noun.

4. In the ethics statement in the Methods and online submission information, please clarify whether consent was written or verbal. 

If verbal, please also specify: i) whether the ethics committee approved the verbal consent procedure, ii) why written consent could not be obtained, and iii) how verbal consent was recorded.

If your study included minors, state whether you obtained consent from parents or guardians.

If the need for consent or parental consent was waived by the ethics committee, please include this information.

5.Thank you for stating the following in the Acknowledgments Section of your manuscript:

'This research project was supported by: the National Geographic Society (Research and Exploration Grant Award# HJ-099R-17), National Science Foundation (IBSS-L Award#1743019), Funding Incentive Seed Grant (University of Utah), the Center for Latin American Studies (University of Utah), Society, Water, & Climate Seed Grant (University of Utah), Nexus Pilot Grant (University of Utah), Economic Science Institute (Chapman University), and the Division of Anthropology (California State University Fullerton).'

'SJM and ES (HJ-099R-17) National Geographic Society Research and Exploration Grant.

SJM (subcontractor 1743019) National Science Foundation IBSS-L (PI Koster) https://www.nsf.gov/awardsearch/showAward?AWD_ID=1743019&HistoricalAwards=false.

The funders had no role in study design, data collection and analysis, decision to publish, or preparation of the manuscript.'

6. We note that Figure 1 in your submission contains map images which may be copyrighted.

We require you to either (a) present written permission from the copyright holder to publish this  figure specifically under the CC BY 4.0 license, or (b) remove the figure from your submission:

b. If you are unable to obtain permission from the original copyright holder to publish this figure under the CC BY 4.0 license or if the copyright holder’s requirements are incompatible with the CC BY 4.0 license, please either i) remove the figure or ii) supply a replacement figure that complies with the CC BY 4.0 license. Please check copyright information on all replacement figures and update the figure caption with source information. If applicable, please specify in the figure caption text when a figure is similar but not identical to the original image and is therefore for illustrative purposes only.

7. Please include captions for your Supporting Information files at the end of your manuscript, and update any in-text citations to match accordingly. Please see our Supporting Information guidelines for more information: http://journals.plos.org/plosone/s/supporting-information

Reviewers' comments:

Reviewer's Responses to Questions

**Comments to the Author**

1. Is the manuscript technically sound, and do the data support the conclusions?

Reviewer #1: Yes

Reviewer #2: Yes

2. Has the statistical analysis been performed appropriately and rigorously? 

Reviewer #1: Yes

Reviewer #2: Yes

3. Have the authors made all data underlying the findings in their manuscript fully available?

Reviewer #1: Yes

Reviewer #2: Yes

4. Is the manuscript presented in an intelligible fashion and written in standard English?

Reviewer #1: Yes

Reviewer #2: Yes

5. Review Comments to the Author

Reviewer #1: This paper sets out to evaluate two commonly held ideas in human demography: first, that dispersal raises age at first marriage and first birth, and second, that education raises age at first marriage and first birth. The authors suggest (with supporting data with respect to dispersal, and a logical argument with respect to education) that dispersal and education may expose individuals to additional marriage partners, especially in small populations where candidate partners are limited. They find evidence for this in Baja California Sur.

This paper has a compelling premise and is largely well-written. The data from BCS are extensive, and I can tell a lot of legwork went into data collection. There are things in the paper that could use some clarification or additional supporting material, however. I’ve provided detailed comments and suggested corrections in the attached PDF. Here, I’ll highlight my main concerns:

• “Exposure” needs to be unpacked a little more here to do justice to the theory. With respect to education especially, there are two ways in which exposure could matter. First, the average person in this population is attending school for about four years. If this is anything like where I work, most people who attend for four years aren’t attending middle school and two years of high school: they’re attending from about age 6-7 until age 10-11. This is possibly far too early for mate search to be taking place. As a robusticity check, I recommend the authors consider treating education as a continuous variable and inspecting whether length of exposure to schooling – and thus length of exposure to that aggregation of potential mates – increases the probability of an individual marrying (or reproducing) early. There are a lot of zeroes for education, I know, but this robusticity check could be run just on participants who attended any school. Second, two of the four communities are “destination” communities when it comes to temporary migration for education, and two are source communities. This should make a big difference with respect to the effect of education. For example, for students coming from a source community to a destination community to attend school, they’re getting exposed to all the students plus all the non-school-attending kids in the community, plus (assuming some age differences between spouses; never established) to potentially marriageable individuals who have already finished school. On the other hand, the kids from the destination community who are staying put are getting exposed to a much smaller pool of potential mates above and beyond those already in their community (especially because the source communities are smaller than the destination communities in this context). As such, I recommend a robusticity check where the authors consider the effects of education separately for educated participants originally from a source community and educated participants originally from a destination community. This is assuming that they know where people were born, of course.

• There was some looseness with the term “population” which seemed to deviate from theoretical work. In the abstract and introduction, the four communities are treated as four different “populations.” At one point in the introduction, the authors discuss populations as having a size of 50-150 people; however, if we’re talking about the classic literature, populations of intermarrying individuals are usually much larger. (I also know that the authors mean “population” in the same way I’m thinking about it – population of interbreeding individuals – as they clarify that communities are not “closed populations” in the methods section.) In the methods section, there is some vagueness about the extent to which these four communities are one intermarrying population (I flagged the relevant sentence with a request for clarification), but over the course of the paper, it’s pretty clear that they are. (The schools aggregate the kids, at the very least, allowing intermarriage.) As such, I recommend the authors stick to the word “community” with respect to the four communities throughout, as they don’t fit the definition of “four populations” as the authors mean “population.” They can clarify that this is a limitation if relevant.

• In the discussion, the authors harness their ethnographic data to talk about the role of education in BSC – and it’s not for getting a degree. However, they don’t do the same in the previous paragraph for dispersal. For example, the authors appear to have data on how a couple met and who moved: the husband or the wife. By their logic, if women are the ones who should be especially likely to marry kin if they disperse, one would expect those cultural systems to be acting for planting women in communities where they have existing kin. Thus, one would expect that among those couples who met through family, the woman would be the one who was most likely to have dispersed to where the man was, rather than vice versa. That ethnographic information would help in the discussion of the dispersal findings.

• I left a comment detailing this, but I was actually a little thrown by the introduction when it came to the predictions of the paper. In the introduction, the authors are careful to distinguish what we already know about the effects of dispersal in small populations from that of what we know about large populations: in large, it delays age at marriage and age at first birth, while among small populations, it appears to do the opposite. They make a similar argument for education: delays in large populations, hastens in small – however, they don’t provide evidence that it hastens in small populations, just make a one-sentence verbal argument. As such, given the overview they provided, I somewhat expected that the predictions would be that dispersal would hasten AFM and AFB and education would delay it, and thus the two would trade off. This was of course wrong on my part, but I want to call the authors’ attention to my misreading in case I’m not the last to get the wrong impression. A little extra flesh on the education paragraph, or the paragraph introducing the predictions, might help alleviate this.

Again, a well-motivated, well-researched, and scientifically sound paper; however, unpacking and checking a few more things would strengthen it further.

Reviewer #2: Overview

In their manuscript titled, “The role of dispersal and school attendance on reproductive dynamics in small, dispersed populations: Choyeros of Baja California Sur, Mexico”, Macfarlan et al. present data from a fascinating study population on dispersal, ages of marriage and first birth, and school attendance. I commend the authors for the hardwork that goes into collecting empirical data like this in a remote small-scale population, and quality of the data appear to be good (also kudos for providing the full analysis dataset). This paper addresses important and understudied questions in human behavioral ecology about the role of dispersal in human mating and alliance formation. The manuscript was well-referenced, well-written and a pleasure to read, the analyses are generally appropriate, and the content fitting to the journal. I look forward to seeing this published and am happy to recommend it for publication.

I have few major critiques, but raise some general and specific comments below.

General comments

1) In general, I think the authors could do more to highlight what is known from the animal literature. For example, circa ~L78, although humans are unique in many ways as noted, we are not the only animals for whom alliance formation and social support are critical. Chimpanzees provide an absolutely fascinating model system for comparison as a male philopatric species that is highly social. Female dispersal is not ubiquitous in chimps, and varies between places. For example, it has been reported that only ~50% of female chimps disperse at Gombe compared to almost 100% in other locations. Pusey et al 1997 have argued that this may be due to differences in the importance of female hierarchies and reproductive skew in these populations.

Unfortunately, there are few studies that have specifically compared the attributes and reproductive outcomes of non-dispersing vs. dispersing female chimps, although a cool paper by Walker et al 2018 recently found that dispersing females have an age at first birth several years later than non-dispersing females. I'm not sure if there are any experimental studies with social rodents or similar that have randomized dispersal to see how it affects outcomes. This literature is large (in contrast to that on humans where the dispersal literature has much to be done), however, and there may be other cases which could highlight the extent to which remaining in one's natal group affects relatedness of mating partners and reproductive parameters.

An additional point is that I think it is not necessarily clear that "individuals who disperse are expected to suffer" -- this is probably a conditional adaptive behavior that does reduce social support temporarily, but which might be regained through affinal kin.

2) Overall, this manuscript frames dispersal as something that is universally good or bad for certain outcomes instead of asking "when/why should an individual disperse?" I realize that the data available might not allow for more in-depth tests at the individual level, but I'm guessing that we could all agree that environmental and individual conditions are likely to govern these decisions at a fine-scale. Perhaps some discussion about condition-dependent dispersal in the broader literature and this study could help bring this issue to the forefront (there is a bit near the end about the effects of wealth citing the papers by Voland which starts to get at this, but the logic is not introduced much).

3) I would generally suggest that the prediction section in the intro (L138-147) could be better framed as questions, rather than specific predictions. Many of these predictions feel tailored to the specific results later, when in reality these are interesting exploratory questions, for example, how does dispersal affect partner genetic relatedness? I leave this up to the authors for consideration, but certainly I don't think that framing it less as specific predictions would reduce the quality or impact of the study.

4) I believe the general conclusions about women marrying more related kin when dispersing is presented too strong. The reality is that we cannot know what the alternative would have been for these specific women (i.e. maybe they would have married even closer kin if staying natally given the available pool, making the dispersal option one of less relatedness). Given that limitation, the main observation is still striking, and it is appropriate to note that even when dispersing there are many marriages to kin occurring (probably for the reasons stated). But the jury will remain out on whether this is a pattern that generalizes.

Specific comments

L37: typo, "may designed in a way"

L37: Phrasing dispersal as being "designed" for one purpose or another might be misleading. It seems more appropriate to say that dispersing females seek partners in novel locales that are likely to elicit strong social support.

L54: Is "similar in age" necessary? Or do you mean sexually mature?

L67: "One solution across…" -- I don't see later that alternative mechanisms are discussed, things like extra-group mating, kin recognition, or delayed maturation. Given that humans have excellent biparental kin recognition given provisioning fathers (doesn't help when no available partners in small group), already have delayed maturation, and extra-group mating raises issues of non bi-parental care, I'm not sure these mechanisms deserve much space, but could be worth considering other options that humans have (polygyny as well).

L72-73: time and risk costs are also highlighted in these excellent citations

L110-112: This is an interesting point, but I think may be stated a bit too forcefully here. I am surprised not to see the paper by Chagnon et al 2017 (https://www.pnas.org/content/114/13/E2590#abstract-2) cited and discussed, given the relevance. I raise this point just to say that different parties involved (e.g. parents vs. offspring) may have different goals for mating partnerships that might not always align. It is also not clear (at least to me) that cross-cousin marriage is linked with severe inbreeding depression to an extent that is easily detectable in outcomes (despite some arguments in the aforementioned linked article). Also see: https://www.nature.com/articles/pr2016177.

L119-120: Unclear to me why this is -- is it because of the point made several lines above that there is greater need for social support?

L132-137: Excellent points. The role of education in limited opportunity environments is understudied despite massive worldwide campaigns to make sure that kids everywhere have access to education (with little regard to what they can do with that education later).

L138-142: predictions 1+3 could be combined? I don't see why males are excluded from prediction 3 though.

-I commend the authors providing a thorough description of study population and environment!

L172: If I recall from one of your other papers, some of the additional colonists (e.g. during the porfiriato period) came from numerous international locations. Does that mean the study population includes a mix of people with backgrounds such as Chinese, Russian, Spanish, etc. as well as indigenous? Are there important lingering differences based on these population histories that determine who marries who?

L234: "household" should be "households"

L241: "reliance of" should be "reliance on"?

-Results: Note that there is some introductory/discussion/methods text mixed into certain areas of the results and does not need to be repeated here.

L281: Interesting that the coefficient for females attending school is nearly as large as that for males, but the difference must be in the variability among individuals leading to a larger standard error. Could it be that females in general have less variability in age at first marriage than males? In that case it would suggest a different interpretation.

-L294: You should make absolutely certain that your statistical software is handling the use of a binomial error distribution correctly. Typically a binomial with logit link function is used in cases where proportions are described by discrete counts of integers (0's and 1's), not for cases where data are inherently proportional. Given that your data are bounded between 0/1 and are inherently proportional, typically a beta distribution would be more appropriate. An issue with a beta regression, however, would arise if you have many datapoints at exactly 0 or 1. In that case a standard transformation (https://stats.stackexchange.com/questions/31300/dealing-with-0-1-values-in-a-beta-regression) or use of a zero/one inflated beta might be necessary. Alternatively a binomial model could be used if the data are split at some interval to represent non-related vs. related.

L305: missing "on" between "attendance age"

Table 4: It seems like dispersal might affect age at first birth via increased age at marriage

L322: Not quite contrary to these findings -- as you note earlier, the costs will vary depending on the reasons for dispersal.

L346-348: Good opportunity to cite Chagnon paper in highlighting parent-offspring conflict in mating decisions.

L357-360: These are interesting ideas, although I don't think the evidence here is definitive. I hope the authors plan to do follow-up studies in this fascinating system to further investigate differences in social support received by mothers who marry kin when dispersing vs those who marry non-kin!

-I am glad to see wealth mentioned in the limitations section, as that was a major question I had until this point. Inheritance in these families is another piece of the puzzle that you probably have ethnographic insight into and which is likely to be particularly important here.

6. PLOS authors have the option to publish the peer review history of their article (what does this mean?). If published, this will include your full peer review and any attached files.

Reviewer #1: No

Reviewer #2: **Yes: **Thomas Kraft

---

## [Author Response · Author response to Decision Letter 0]

18 Aug 2020

Reference: Manuscript ID PONE-D-20-06987

Title: The role of dispersal and school attendance on reproductive dynamics in small, dispersed populations: Choyeros of Baja California Sur, Mexico

Response to Reviewers

We thank the editor and the two reviewers for their insightful and necessary suggestions to our MS. As a result, we have substantially altered the manuscript to address these issues. Below, you will find detailed responses directed to each reviewers’ specific commentary. 

Editor’s Comment 1. Eliminate causal language when your results only imply statistical relationships. This applies throughout the manuscript. For example, in the abstract: “For men, dispersal results in younger ages of marriage” – should become “For men, dispersal is associated with younger ages of marriage”. 

Authors’ Response: Excellent point and we agree completely. We have made these changes throughout the MS including the Abstract, Results, and Discussion sections.

2. While you do discuss limitations at the end of the manuscript. I would like to see a more dedicated, upfront acknowledgement and discussion of the limitations of your methodology specifically with respect to isolating causality. My concerns here are not so much that causality might be reversed, but rather that the associations you report are very likely also associated with confounding 3rd variables. There is a general sense in which you treat going to school and dispersal as almost experimental factors, but a large body of demographic research tells us that migrants are rarely comparable to those who do not migrate. Similarly, those that go to school likely come from families of distinct socioeconomic backgrounds that those who do not. Therefore, your manuscript must more clearly address which individuals are most likely to disperse and go to school and discuss the role of potential confounds. The reality is that while you have a lot of ethnographic knowledge to draw on, and a very sophisticated grasp of the theory and past literature- the statistical models are very simplistic and the analysis dangerously underpowered - especially by the standards of demography. I realize these limitations are balanced by the novelty of the data and fieldwork commitments, but they need to be more clearly acknowledged. 

Authors’ Response: This is completely reasonable. We have now included statements throughout the manuscript that 1) deals with our inability to detect causality and the role of potential confounds (in the “Analysis” sub-section as well as in the “Discussion” section); and 2) factors causing variability in education and dispersal (in the “Study Site” and “Discussion” sections, as well as the “Dispersal, marriage, and reproductive data” sub-section). 

3. A concise mention of these limitations should be made in the abstract itself. The abstract should also mention the sample size.

Authors’ Response: We now include a statement about the limitations of statistical tests based on small sample sizes in the Abstract and include the sample size.

4. Linking to the above points, reading the manuscript I was not entirely clear of the extent to which dispersal and education are overlapping phenomena for men and women. How many folks go to school but don't disperse and vice-versa. This needs to be made more clear. 

Authors’ Response: This is a good point. We have now included a simple Chi Square analysis in the “Data Section” labeled “Dispersal, marriage, and reproductive data” that shows that individuals who attended school were less likely to disperse. We state the following: 

“A relationship existed between education and dispersal, such that individuals who had attended school were less likely to disperse relative to those with no education (X2=4.8; p=.03; n=104 (Natal-Education n=47; Dispersed-Education n=33; Natal-No Education n=8; Dispersed-No Education n=16).”

Reviewers' comments:

Reviewer's Responses to Questions

Reviewer #1: This paper sets out to evaluate two commonly held ideas in human demography: first, that dispersal raises age at first marriage and first birth, and second, that education raises age at first marriage and first birth. The authors suggest (with supporting data with respect to dispersal, and a logical argument with respect to education) that dispersal and education may expose individuals to additional marriage partners, especially in small populations where candidate partners are limited. They find evidence for this in Baja California Sur.

This paper has a compelling premise and is largely well-written. The data from BCS are extensive, and I can tell a lot of legwork went into data collection. There are things in the paper that could use some clarification or additional supporting material, however. I’ve provided detailed comments and suggested corrections in the attached PDF. Here, I’ll highlight my main concerns:

Reviewer’s Comment: “Exposure” needs to be unpacked a little more here to do justice to the theory. With respect to education especially, there are two ways in which exposure could matter. First, the average person in this population is attending school for about four years. If this is anything like where I work, most people who attend for four years aren’t attending middle school and two years of high school: they’re attending from about age 6-7 until age 10-11. This is possibly far too early for mate search to be taking place. As a robusticity check, I recommend the authors consider treating education as a continuous variable and inspecting whether length of exposure to schooling – and thus length of exposure to that aggregation of potential mates – increases the probability of an individual marrying (or reproducing) early. There are a lot of zeroes for education, I know, but this robusticity check could be run just on participants who attended any school. 

Authors’ Response: We agree with the reviewer on a number of points they raise. Yes, in this population, children typically only attend primary school (aka “Primaria”: grades 1-6). And yes, children of this age-set typically are not in the marriage market. However, interviews and conversations with community members led us to conclude that what matters most for locating marriage partners (or even friends) is simply having any exposure to them, especially people from other communities. As they told us, being aware that people (especially members of the opposite sex) simply existed became the pretext for seeing them (and communicating with them) in the future. This was the case, whether it was a rancher with no education looking for their goats in another community or a person who attended school for 5 years. From what we can tell, schools in these locations accelerate how quickly people develop mental models of where others reside and how they might allocate their time to extend their social networks. By having exposure to a member of the opposite sex via school, this meant that they could more easily initiate a conversation with them at festivals, parties, and other events that bring people together. 

 In an effort to comply the reviewer’s comments, we examined the effect of years of education on marriage and reproductive outcomes for those who had any schooling. These models are potentially underpowered (Male n=37; Female n=42). We find the following relationships using the same statistical models as presented in the MS (i.e., Generalized Estimating Equation with Robust Standard Errors using an Independent Correlation Structure around Community ID; Independent Variables associated with the analysis “Age at Marriage” include “Year of Marriage”, “Dispersal”, and “Education Years”; Independent Variables associated with “Age at First Birth” include: “Age at Marriage”, “Dispersal”, and “Years of Education”).

a) No relationship exists between Years of Education and Male Age at Marriage.

b) No relationship exists between Years of Education and Female Age at Marriage; 

c) No relationship exists between Age at First Birth and Years of Education. 

Given our understanding of the ethnographic context, in conjunction with the analytic models, we feel assured using our initial variable “Groom/Bride Ever Attended School”. 

Reviewer Comment: Second, two of the four communities are “destination” communities when it comes to temporary migration for education, and two are source communities. This should make a big difference with respect to the effect of education. For example, for students coming from a source community to a destination community to attend school, they’re getting exposed to all the students plus all the non-school-attending kids in the community, plus (assuming some age differences between spouses; never established) to potentially marriageable individuals who have already finished school. On the other hand, the kids from the destination community who are staying put are getting exposed to a much smaller pool of potential mates above and beyond those already in their community (especially because the source communities are smaller than the destination communities in this context). As such, I recommend a robusticity check where the authors consider the effects of education separately for educated participants originally from a source community and educated participants originally from a destination community. This is assuming that they know where people were born, of course.

Authors’ Responses: These are really interesting points, especially the concepts of destination versus source communities. With respect to the idea that all individuals in a “destination” community are equally exposed to children from source communities requires some clarification. At face value, it seems like a reasonable assumption; however, the ethnographic reality is different. We are aware of a number of children, who currently reside in a community with a school, however, these children never attend it because the distances are too far or too difficult to travel. So, there are real differences in exposure within a destination community. Individuals from households immediately adjacent to a school will have greater exposure, but not necessarily people who reside far away from the school (e.g. 20+ km away). 

Unfortunately, we are unable to perform the robusticity test (i.e., the effects of education separately for educated participants originally from a source community and educated participants originally from a destination community) suggested by the reviewer, as 1) we lack data on where individuals obtained their education for those who migrated into one of the four communities; and 2) the sample sizes are really small. 

Last, we now include a line on Table 1 where we provide Spousal Age Difference (Groom’s Age – Bride’s Age): Mean = 5-years; Median = 5-years; Min/Max= -9/21 years; n=50.

Reviewer Comment: There was some looseness with the term “population” which seemed to deviate from theoretical work. In the abstract and introduction, the four communities are treated as four different “populations.” At one point in the introduction, the authors discuss populations as having a size of 50-150 people; however, if we’re talking about the classic literature, populations of intermarrying individuals are usually much larger. (I also know that the authors mean “population” in the same way I’m thinking about it – population of interbreeding individuals – as they clarify that communities are not “closed populations” in the methods section.) In the methods section, there is some vagueness about the extent to which these four communities are one intermarrying population (I flagged the relevant sentence with a request for clarification), but over the course of the paper, it’s pretty clear that they are. (The schools aggregate the kids, at the very least, allowing intermarriage.) As such, I recommend the authors stick to the word “community” with respect to the four communities throughout, as they don’t fit the definition of “four populations” as the authors mean “population.” They can clarify that this is a limitation if relevant.

Authors’ Response: This is a useful critique and we have adjusted the manuscript accordingly. We now refer to the four groups as “communities” not “populations”. However, we retain the use of population when discussing previous literature on the demography of small populations, as well as on the meta-population of ranchers in BCS, Mexico.

Reviewer Comment: In the discussion, the authors harness their ethnographic data to talk about the role of education in BSC – and it’s not for getting a degree. However, they don’t do the same in the previous paragraph for dispersal. For example, the authors appear to have data on how a couple met and who moved: the husband or the wife. By their logic, if women are the ones who should be especially likely to marry kin if they disperse, one would expect those cultural systems to be acting for planting women in communities where they have existing kin. Thus, one would expect that among those couples who met through family, the woman would be the one who was most likely to have dispersed to where the man was, rather than vice versa. That ethnographic information would help in the discussion of the dispersal findings.

Authors’ Response: This is another interesting point. We wish we had the sample sizes to adjudicate this relationship in a satisfying manner. However, amongst the eight couples who were introduced to each other by family members, five involved marriages to genetic relatives. In each of these five cases, the female dispersed from her natal community. 

Reviewer Comment: I left a comment detailing this, but I was actually a little thrown by the introduction when it came to the predictions of the paper. In the introduction, the authors are careful to distinguish what we already know about the effects of dispersal in small populations from that of what we know about large populations: in large, it delays age at marriage and age at first birth, while among small populations, it appears to do the opposite. They make a similar argument for education: delays in large populations, hastens in small – however, they don’t provide evidence that it hastens in small populations, just make a one-sentence verbal argument. As such, given the overview they provided, I somewhat expected that the predictions would be that dispersal would hasten AFM and AFB and education would delay it, and thus the two would trade off. This was of course wrong on my part, but I want to call the authors’ attention to my misreading in case I’m not the last to get the wrong impression. A little extra flesh on the education paragraph, or the paragraph introducing the predictions, might help alleviate this.

Authors’ Response: Thank you for pointing this out. We have revised this paragraph accordingly. 

Author Comments to additional reviewer commentary in the PDF document

Reviewer Comment: Did you center or subtract the minimum to avoid generating estimates for ages younger than 14 and thus affecting estimated slope?

Authors’ Response: We did not center our data nor did we subtract the minimum value for our variables. Our analytic philosophy is to not transform variables unless necessary for interpretability. We feel for the flow of the manuscript and readability that we are better served by keeping the variables in their “natural state”. However, we did perform this data transformation and re-ran the associated analyses to evaluate any doubts. The results are substantively identical.

Reviewer Comments: Although looking at the sex ratio, it seems like a chunk of the females are migrating all the way out of this population of four communities, and these communities do have exposure to towns through markets... no one's moving there...?

Authors’ Response: You are correct. Migration is definitely taking place to urban environments; however, it occurs at a lower frequency than remaining in the rural ranching communities. But, it appears very likely that the male sex-biased ratios are being driven by female dispersal to urban environments. We did not include this dataset in this manuscript for fear that it might confuse readers (1 dataset dealing with adults who have married – which is the focus of our manuscript; another dataset focusing largely on the children of those who married; with a small overlap between datasets). During our census/interviews we asked heads of households to provide information on all their children, including those who have moved away. A sample of 65 households that had children and who provided information on their whereabouts (and their education achievement) provided us information on 214 children (109 female; 105 male). A number of these children are too young to have left home yet. If we, filter the data to include just those children who are aged 16 or older this provides a sample of 150 children (75 males; 75 females). Fifty-five of these children (37%; 33 females, 22 males) dispersed to an urban locale and 95 (63%; 42 females; 53 males) remained within the rural mountain range either in their natal home, their natal community, or another ranching community. Given this information, we now include a statement in the “Data” section that states the following:

“The four communities all demonstrate male-biased sex ratios at the time of data collection (mean=1.2; range: 1.07 to 1.8) and ethnographic interviews suggest this is largely driven by female dispersal to urban environments.”

 

Reviewer #2: Overview

In their manuscript titled, “The role of dispersal and school attendance on reproductive dynamics in small, dispersed populations: Choyeros of Baja California Sur, Mexico”, Macfarlan et al. present data from a fascinating study population on dispersal, ages of marriage and first birth, and school attendance. I commend the authors for the hardwork that goes into collecting empirical data like this in a remote small-scale population, and quality of the data appear to be good (also kudos for providing the full analysis dataset). This paper addresses important and understudied questions in human behavioral ecology about the role of dispersal in human mating and alliance formation. The manuscript was well-referenced, well-written and a pleasure to read, the analyses are generally appropriate, and the content fitting to the journal. I look forward to seeing this published and am happy to recommend it for publication.

I have few major critiques, but raise some general and specific comments below.

General comments

1) In general, I think the authors could do more to highlight what is known from the animal literature. For example, circa ~L78, although humans are unique in many ways as noted, we are not the only animals for whom alliance formation and social support are critical. Chimpanzees provide an absolutely fascinating model system for comparison as a male philopatric species that is highly social. Female dispersal is not ubiquitous in chimps, and varies between places. For example, it has been reported that only ~50% of female chimps disperse at Gombe compared to almost 100% in other locations. Pusey et al 1997 have argued that this may be due to differences in the importance of female hierarchies and reproductive skew in these populations.

Unfortunately, there are few studies that have specifically compared the attributes and reproductive outcomes of non-dispersing vs. dispersing female chimps, although a cool paper by Walker et al 2018 recently found that dispersing females have an age at first birth several years later than non-dispersing females. I'm not sure if there are any experimental studies with social rodents or similar that have randomized dispersal to see how it affects outcomes. This literature is large (in contrast to that on humans where the dispersal literature has much to be done), however, and there may be other cases which could highlight the extent to which remaining in one's natal group affects relatedness of mating partners and reproductive parameters.

An additional point is that I think it is not necessarily clear that "individuals who disperse are expected to suffer" -- this is probably a conditional adaptive behavior that does reduce social support temporarily, but which might be regained through affinal kin.

Authors’ Response: We appreciate the reviewer’s (Tom’s) concern here. We have cited a representative sample of relevant research from both the human and non-human animal literatures [including one paper that synthesizes research on dispersal on over 257 species (Trochet et al. 2016) for which we forgot to include the in-text citation – sorry about that]. Based on our reading of the literature (which is by no means exhaustive), two trends emerge: a) it is generally presented as detrimental to the individual who disperses (which is what we are challenging) and b) there is debate regarding whether dispersal is driven by inbreeding avoidance versus other mechanisms (e.g., resource scarcity, mate competition). Our goal is not to synthesize this entire literature, rather we seek to acknowledge that these intellectual traditions exist and use them to help frame our perspective. We now cite the research by Walker et al. (2018). 

2) Overall, this manuscript frames dispersal as something that is universally good or bad for certain outcomes instead of asking "when/why should an individual disperse?" I realize that the data available might not allow for more in-depth tests at the individual level, but I'm guessing that we could all agree that environmental and individual conditions are likely to govern these decisions at a fine-scale. Perhaps some discussion about condition-dependent dispersal in the broader literature and this study could help bring this issue to the forefront (there is a bit near the end about the effects of wealth citing the papers by Voland which starts to get at this, but the logic is not introduced much).

Authors’ Response: We are amenable to this position, specifically as it relates to humans. However, for many other species with sex-specific dispersal, this would not be relevant. We now include a few sentences in the Discussion dedicated to highlighting that environmental and individual conditions affect decisions to migrate. We now state the following:

“While dispersal is often presented as being either uniformly negative or positive on individual outcomes, a more nuanced approach is likely appropriate given that decisions to migrate are conditioned by environmental and/or individual-level variability, such as local mate scarcity, habitat suitability and saturation, and kinship institutions. We plan to target future research on the multiple motivations for and consequences of dispersal on individual outcomes.”

3) I would generally suggest that the prediction section in the intro (L138-147) could be better framed as questions, rather than specific predictions. Many of these predictions feel tailored to the specific results later, when in reality these are interesting exploratory questions, for example, how does dispersal affect partner genetic relatedness? I leave this up to the authors for consideration, but certainly I don't think that framing it less as specific predictions would reduce the quality or impact of the study.

Authors’ Response: We make this edit and refer to them as questions that we examine.

4) I believe the general conclusions about women marrying more related kin when dispersing is presented too strong. The reality is that we cannot know what the alternative would have been for these specific women (i.e. maybe they would have married even closer kin if staying natally given the available pool, making the dispersal option one of less relatedness). Given that limitation, the main observation is still striking, and it is appropriate to note that even when dispersing there are many marriages to kin occurring (probably for the reasons stated). But the jury will remain out on whether this is a pattern that generalizes.

Authors’ Response: We agree that we were too cavalier in how we presented our findings. We do not know what the pool of alternative marriage partners looked like. We have tempered our wording throughout the MS, to indicate that our findings are simply statistical associations. 

Specific comments

L37: typo, "may designed in a way"

L37: Phrasing dispersal as being "designed" for one purpose or another might be misleading. It seems more appropriate to say that dispersing females seek partners in novel locales that are likely to elicit strong social support.

Authors’ Response: We have removed the wording about “design” and state: 

“This finding suggests that human dispersal may promote female social support from genetic kin in novel locales for raising offspring.”

L54: Is "similar in age" necessary? Or do you mean sexually mature?

Authors’ Response: We meant similar in age, since there is considerable cross-cultural evidence that people prefer marriage partners who are generally similar in age (+/- 5 years). However, we are happy to remove it for space and clarity.

L67: "One solution across…" -- I don't see later that alternative mechanisms are discussed, things like extra-group mating, kin recognition, or delayed maturation. Given that humans have excellent biparental kin recognition given provisioning fathers (doesn't help when no available partners in small group), already have delayed maturation, and extra-group mating raises issues of non bi-parental care, I'm not sure these mechanisms deserve much space, but could be worth considering other options that humans have (polygyny as well).

Authors’ Response: While we appreciate the reviewer’s concerns and it is clear that alternative mechanisms might be applicable (depending on the species), we would prefer to keep our focus on the few items that our manuscript actually can speak to (namely, dispersal and education). 

L72-73: time and risk costs are also highlighted in these excellent citations

Authors Response: Agreed

L110-112: This is an interesting point, but I think may be stated a bit too forcefully here. I am surprised not to see the paper by Chagnon et al 2017 (https://www.pnas.org/content/114/13/E2590#abstract-2) cited and discussed, given the relevance. I raise this point just to say that different parties involved (e.g. parents vs. offspring) may have different goals for mating partnerships that might not always align. It is also not clear (at least to me) that cross-cousin marriage is linked with severe inbreeding depression to an extent that is easily detectable in outcomes (despite some arguments in the aforementioned linked article). Also see: https://www.nature.com/articles/pr2016177.

Authors’ Response: We have now included the Chagnon et al. (2017) citation. And we agree that there is likely conflict between parents and children regarding mating decisions. However, here we target married adults and so have no data on their children. That said, we appreciate this idea as a question for research – thank you!

L119-120: Unclear to me why this is -- is it because of the point made several lines above that there is greater need for social support?

Authors’ Response: Yes, we are making this prediction (or posing this question) because 1) the animal ethology literature largely paints dispersal as a way to dampen inbreeding, 2) social support from genetic kin is really important to humans, and 3) humans engage in cross-cousin marriage using social institutions to facilitate the movement of people into new communities. Point 1 does not square with point 3 and many researchers (who are unware of point 3) believe that human dispersal dampens inbreeding. One of the novel components of our manuscript (we hope), is that we are drawing attention to these contradictions. Human dispersal (in small, dispersed populations), might be very different from large human societies, as well as other species.

L132-137: Excellent points. The role of education in limited opportunity environments is understudied despite massive worldwide campaigns to make sure that kids everywhere have access to education (with little regard to what they can do with that education later).

Authors’ Response: Awesome! Thanks!

L138-142: predictions 1+3 could be combined? I don't see why males are excluded from prediction 3 though.

Author’s Response: While I assume that predictions 1 and 3 could be combined, for consistency in how we present our results, we decided to separate them out. We did not include males in prediction 3 following conventions in demographic research typically only examining reproductive outcomes on females

-I commend the authors providing a thorough description of study population and environment!

Authors’ Response: Awesome! Thanks. We had fun writing it.

L172: If I recall from one of your other papers, some of the additional colonists (e.g. during the porfiriato period) came from numerous international locations. Does that mean the study population includes a mix of people with backgrounds such as Chinese, Russian, Spanish, etc. as well as indigenous? Are there important lingering differences based on these population histories that determine who marries who?

Authors’ Response: Based on the genealogies we have collected and the interviews we’ve performed, the population descends from two colonization events (brought with the Jesuits during 17th and 18th century, or after Mexican Independence). None of the population claims ancestry based on Chinese, Russian, or Indigenous peoples (the entire indigenous population in this region, the Guyacura, were forcibly relocated to the Cape Region of southern Baja California Sur in 1768). All four of these communities (as well as others in the Giganta mountain range) identify each other as being part of the same ethnic group: Choyero or Ranchero (Choyero is a regional variant of Ranchero). No one from the region has ever expressed that alternative ethnicities reside within the mountain range. We feel confident in saying that there are no meaningful ethnic differences that would affect marriages in this region.

L234: "household" should be "households"

Authors’ Response: Done

L241: "reliance of" should be "reliance on"?

Authors’ Response: Done

-Results: Note that there is some introductory/discussion/methods text mixed into certain areas of the results and does not need to be repeated here.

Authors’ Response: Although we would prefer to include a little background information within the context of the Results’ research questions to remind the reader the motivation for our study, we have modified these paragraphs to suit this request. 

L281: Interesting that the coefficient for females attending school is nearly as large as that for males, but the difference must be in the variability among individuals leading to a larger standard error. Could it be that females in general have less variability in age at first marriage than males? In that case, it would suggest a different interpretation.

Authors’ Response: The variability in age at marriage by sex is not significantly different (f=.94; p=.6; n=103) 

Females: n=53; Mean(SD)=21.6(5.9); SE=0.81 

Males: n=50; Mean(SD)=25.9(6.1); SE=0.86

-L294: You should make absolutely certain that your statistical software is handling the use of a binomial error distribution correctly. Typically a binomial with logit link function is used in cases where proportions are described by discrete counts of integers (0's and 1's), not for cases where data are inherently proportional. Given that your data are bounded between 0/1 and are inherently proportional, typically a beta distribution would be more appropriate. An issue with a beta regression, however, would arise if you have many datapoints at exactly 0 or 1. In that case a standard transformation (https://stats.stackexchange.com/questions/31300/dealing-with-0-1-values-in-a-beta-regression) or use of a zero/one inflated beta might be necessary. Alternatively a binomial model could be used if the data are split at some interval to represent non-related vs. related.

Authors’ Response: We appreciate this comment! Given the nature of the data (a fraction ranging between 0 and 1 with data points at 0), we have opted to rerun the analysis using Fractional Regression. Despite the change in statistical modeling, our results still hold. The relevant sections of the manuscript (Analysis and Results) have been adjusted to reflect these changes.

L305: missing "on" between "attendance age"

Authors Response: Done

Table 4: It seems like dispersal might affect age at first birth via increased age at marriage

Authors’ Response: While this might be the case, the statistical model does not corroborate it (the coefficient for Dispersal is not significant; however, our analysis is underpowered, so it could be present, but at a small effect size). Furthermore, we do not have strong evidence that dispersal decreases age at marriage for females (Table 2)

L322: Not quite contrary to these findings -- as you note earlier, the costs will vary depending on the reasons for dispersal.

Authors Response: We agree with what you say, but the papers we cite report that dispersal is associated with later ages at marriage and reproduction in females. Our analyses do not corroborate the findings in these papers. 

L346-348: Good opportunity to cite Chagnon paper in highlighting parent-offspring conflict in mating decisions.

Authors’ Response: Done

L357-360: These are interesting ideas, although I don't think the evidence here is definitive. I hope the authors plan to do follow-up studies in this fascinating system to further investigate differences in social support received by mothers who marry kin when dispersing vs those who marry non-kin!

Authors’ Response: We do plan on following this research thread in the near future. Thanks for the encouragement!

-I am glad to see wealth mentioned in the limitations section, as that was a major question I had until this point. Inheritance in these families is another piece of the puzzle that you probably have ethnographic insight into and which is likely to be particularly important here.

---

## [Editor Report · Decision Letter 1]

9 Sep 2020

The role of dispersal and school attendance on reproductive dynamics in small, dispersed populations: Choyeros of Baja California Sur, Mexico

PONE-D-20-06987R1

Dear Dr. Macfarlan,

We’re pleased to inform you that your manuscript has been judged scientifically suitable for publication and will be formally accepted for publication once it meets all outstanding technical requirements. You have done a solid job on the revisions. Thanks for engaging with each criticism wholeheartedly - the manuscript has much improved. Please also accept my apologies on behalf of the journal for the various delays you have experienced during the editorial process. These are unusual times. 

Kind regards,

David W Lawson

Academic Editor

PLOS ONE

---

## [Editor Report · Acceptance letter]

15 Sep 2020

PONE-D-20-06987R1 

The role of dispersal and school attendance on reproductive dynamics in small, dispersed populations: *Choyeros* of Baja California Sur, Mexico 

Dear Dr. Macfarlan:

I'm pleased to inform you that your manuscript has been deemed suitable for publication in PLOS ONE. Congratulations! Your manuscript is now with our production department. 

Kind regards, 

on behalf of

Dr. David W Lawson 

Academic Editor

PLOS ONE